# Light-steered locomotion of muscle-like hydrogel by self-coordinated shape change and friction modulation

Qing Li Zhu[1], Cong Du[1], Yahao Dai[1], Matthias Daab[2], Marian Matejdes[2], Josef Breu [2], Wei Hong [3✉], Qiang Zheng[1] & Zi Liang Wu [1✉]

Many creatures have the ability to traverse challenging environments by using their active muscles with anisotropic structures as the motors in a highly coordinated fashion. However, most artificial robots require multiple independently activated actuators to achieve similar purposes. Here we report a hydrogel-based, biomimetic soft robot capable of multimodal locomotion fueled and steered by light irradiation. A muscle-like poly(N-isopropylacrylamide) nanocomposite hydrogel is prepared by electrical orientation of nanosheets and subsequent gelation. Patterned anisotropic hydrogels are fabricated by multi-step electrical orientation and photolithographic polymerization, affording programmed deformations. Under light irradiation, the gold-nanoparticle-incorporated hydrogels undergo concurrent fast isochoric deformation and rapid increase in friction against a hydrophobic substrate. Versatile motion gaits including crawling, walking, and turning with controllable directions are realized in the soft robots by dynamic synergy of localized shape-changing and friction manipulation under spatiotemporal light stimuli. The principle and strategy should merit designing of continuum soft robots with biomimetic mechanisms.

[1] Ministry of Education Key Laboratory of Macromolecular Synthesis and Functionalization, Department of Polymer Science and Engineering, Zhejiang University, Hangzhou 310027, China. [2] Bavarian Polymer Institute and Department of Chemistry, University of Bayreuth, Universitätsstrasse 30, Bayreuth 95440, Germany. [3] Department of Mechanics and Aerospace Engineering, Southern University of Science and Technology, Shenzhen 518055, China. ✉email: hongw@sustech.edu.cn; wuziliang@zju.edu.cn

Many living organisms have the ability to navigate through complicated environments by crawling, swimming, flying, and so forth[1–3]. Musculature serves as the basic motor and plays an essential role in these motions by working in a highly coordinated fashion[1]. For example, jellyfish swims by repetitive contracting sub-umbrella striated muscles to expel water and generate propulsion. Worms rely on single or multiple muscular hydrostats, and accomplish crawling by coupled radial contraction and axial elongation of the body[2]. Inspired by these biological paradigms, researchers have developed myriad soft actuators based on soft active materials[4–13]. Hydrogels are recognized as an ideal material for soft actuators and robots because of their similarity to soft biotissues and responsiveness to diverse stimuli[14–19].

Shape change of hydrogel-based actuators usually relies on structural heterogeneity or asymmetric stimulation, and the solvent-diffusion-controlled responses are usually slow[19]. Directional locomotion has been realized by using a ratchet mechanism[14,15,20] or an asymmetrically shaped hydrogel with reversible deformation[21,22]. For instance, autonomous walking is realized in a self-oscillating hydrogel, where mechanical oscillation (i.e. reversible bending and stretching) is converted to directional locomotion of the hydrogel on a ratchet substrate[15]. However, bidirectional or omnidirectional locomotion, which is skillfully mastered by living organisms through coordinated actuation and distributed friction, remains a grand challenge in artificial systems.

Among all methods of locomotion for soft robots, light actuation is particularly attractive, because the non-contact means of manipulation with high spatiotemporal resolution affords flexibility in motile direction and mode[20,23–28]. In addition, muscle-like hydrogels with anisotropic structure and fast response should favor the programmed deformation/locomotion of the soft robots, although the material design has been a long-standing challenge. Recently, Aida and co-workers[29,30] have developed a monodomain poly(N-isopropylacrylamide) (PNIPAAm) hydrogel by using a super-strong magnetic field to align the titanate nanosheets (NSs). After incorporating gold nanoparticles (AuNPs), this hydrogel showed earthworm-like peristaltic crawling in a confined glass tube upon periodic light irradiation, which resulted in radial contraction and longitudinal extension of the cylindrical gel and thus different contact with the tube for translational motion[30]. However, it remains challenging to construct elaborate hydrogel structures for swift locomotion in a free space without geometric constraints.

Here we demonstrate the fabrication of composite hydrogels with intricate ordered structures by sequential electric-field-induced orientation of NSs and photolithographic polymerization of a precursor solution containing NSs and AuNPs. The monodomain hydrogels exhibit rapid, reversible, and isochoric anisotropic deformation upon heating or light irradiation. The patterned anisotropic hydrogels with programmed internal stresses deform into various three-dimensional (3D) configurations. Versatile modes of locomotion, including crawling, walking, and turning, with controllable directions are realized in these muscle-like hydrogels by sweeping a green-light beam through the hydrogel at proper directions and rates. The excellent control of locomotion relies on the spatiotemporal actuation, mismatched deformation, and dynamic control of friction in a coordinated manner. We envision that the approach to fabricate soft materials with ordered structures and the strategy to realize locomotion by asymmetric actuation are applicable to other systems and may facilitate versatile design of soft robots.

## Results

### Synthesis, structure, and properties of monodomain hydrogel.
The fluorohectorite $[Na_{0.5}][Li_{0.5}Mg_{2.5}][Si_4]O_{10}F_2$ NSs used have a

high aspect ratio of ~20,000 and a charge density of $1.1\ nm^{-2}$, enabling repulsive osmotic delamination into single lamellae in water (Supplementary Fig. 1)[31,32]. The aqueous suspension of NSs forms a homogeneous liquid crystalline phase at a very low content, $C_{NS}$, of 0.3 wt%. The multi-domain mesophase can be aligned by mechanical shearing (Supplementary Fig. 2); the correlation between orientation and birefringence under a polarizing optical microscope (POM) provides a simple way to characterize the alignment of NSs[33].

A high-frequency alternating current (AC) electric field is applied to orient the NSs and avoid the electrophoresis of NSs and the electrolysis of water[34]. After applying the electric field for 60 min, the suspension shows strong and uniform birefringence, indicating the unidirectional alignment of NSs along the electric field (Supplementary Fig. 3). This is because of the anisotropic electric susceptibility and dielectric constants in the normal and plane directions of NSs (Supplementary Fig. 4)[21,35]. Systematic experiments are performed to optimize the synthesis parameters for electrical orientation (Supplementary Fig. 5), which are set as a $C_{NS}$ of 0.3 wt%, an electrode distance of 20 mm, a voltage amplitude of 50 V, a frequency of 10 kHz, and a field action time of 60 min in the following sections.

Compared with the super-strong magnetic field, the low-voltage electric field used to align NSs has several advantages. (i) The set-up is low-cost, easy to assemble, and safe to manipulate. (ii) It is facile to tune the distribution of electric field by modulating the electrodes. (iii) It enables the simultaneous application of other techniques such as photolithography or 3D printing[19,36]. Therefore, it paves the way to construct muscle-like hydrogels and biomimetic soft robots, as described as follows.

The cofacially oriented NSs can be permanently embedded in a gel matrix. As shown in Fig. 1a, b, the presence of monomer, crosslinker, and initiator does not influence the electrical orientation of NSs. After the external field is switched off, the alignment of NSs in the precursor is well maintained within 8 min, but deteriorates gradually after long-term storage due to structural relaxation (Supplementary Fig. 6). A monodomain hydrogel is obtained by subsequent photo-polymerization of the aligned precursor within 2 min. After immersed in water at room temperature, the resultant PNIPAAm hydrogel exhibits slightly anisotropic swelling behavior. The linear swelling ratios, $S$, defined as the length ratio of the gel in the equilibrated state to the as-prepared state, in the directions parallel and perpendicular to the alignment of NSs are 1.2 and 1.3, respectively.

To characterize the anisotropic structure, the hydrogel is examined under POM. Besides the observation from the top of the gel sheet (defined as the normal direction, $n$), it is also inspected from the lateral directions parallel and perpendicular to the alignment of NSs (defined as the // and ⊥ directions, respectively). Strong and uniform birefringence is observed from the $n$ and // directions, whereas no birefringence is seen from the ⊥ direction (Fig. 1c and Supplementary Fig. 7), indicating that the NSs align cofacially along the electric field and homeotropically on the substrate of the reaction cell. The alignment of NSs is also confirmed by small-angle X-ray scattering (SAXS) measurements; strong anisotropic scattering is observed in the $n$ and // directions rather than in the ⊥ direction of the hydrogel (Fig. 1d, e). Accordingly, the plane-to-plane distance and orientation dergee of NSs within the as-prepared and equilibrated gels are estimated to be 11.6 nm, 0.88 and 13.4 nm, 0.84, respectively (Supplementary Fig. 8). The highly ordered alignment of NSs is attributed to the high charge density and anisotropic dielectric constant; under an electric field, NSs align cofacially to maximize the electronic repulsion[21,35].

The monodomain hydrogel shows anisotropic mechanical properties, with Young's modulus along the // direction ~3 times

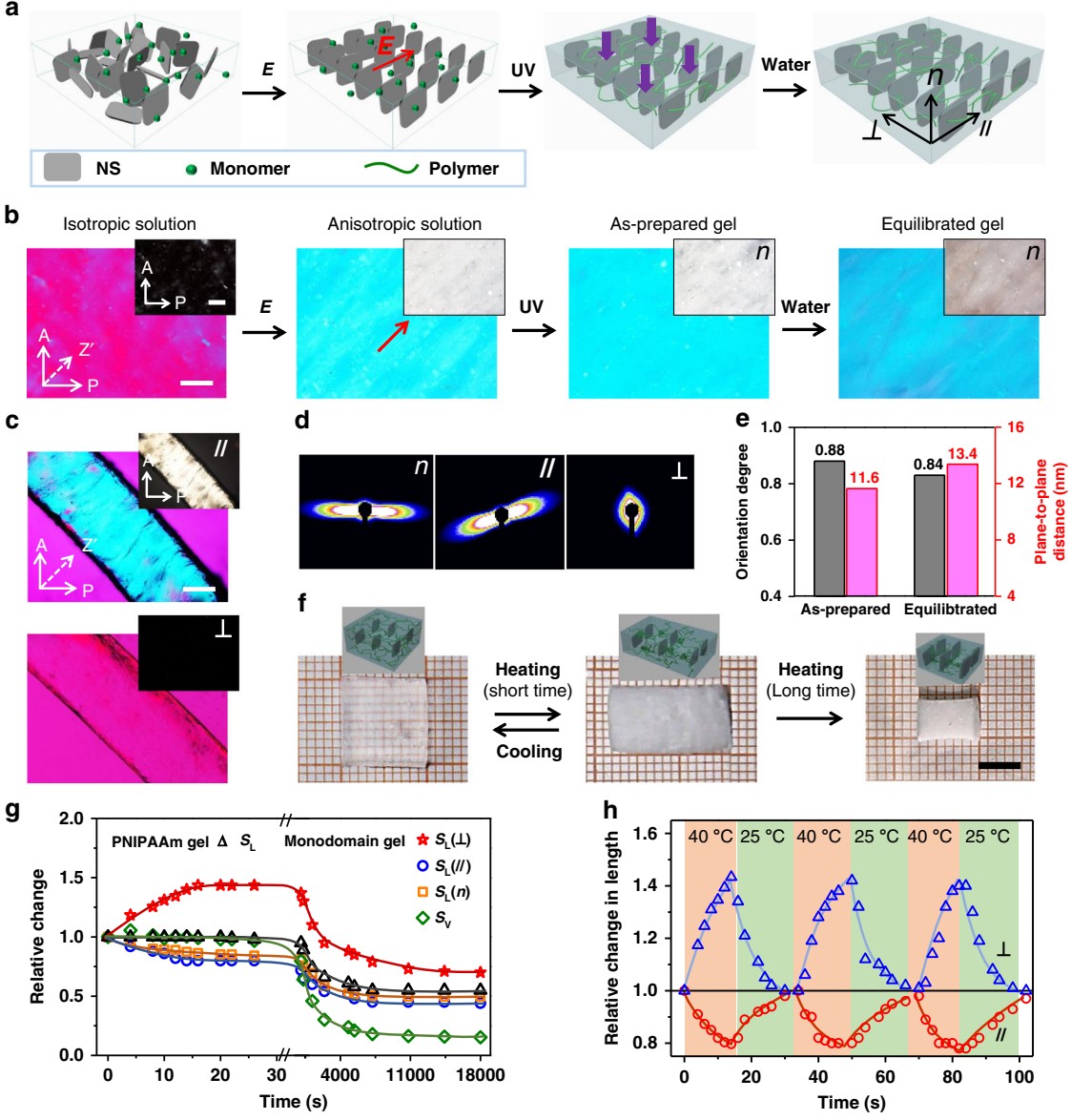

**Fig. 1 Synthesis, structure, and properties of the monodomain hydrogel. a, b** Schematic for the preparation of monodomain anisotropic hydrogel by electric-field-induced orientation of NSs (**a**) and corresponding polarizing optical microscope (POM) images of the sample viewed from the top (**b**). A: analyzer, P: polarizer, Z′: slow axis of the 530 nm tint plate. **c–e** POM images of the sample observed from the // and ⊥ directions (**c**), 2D small-angle X-ray scattering patterns (**d**), and corresponding plane-to-plane distance and orientation dergee of NSs in the gel (**e**). **f** Photos and structure illustrations to show the temperature-mediated anisotropic deformation of the hydrogel. **g, h** Variations of the dimensions of hydrogel after transferring it from 25 to 40 °C water bath for a long time (**g**) or cyclically switching between 25 and 40 °C water baths with a short interval (**h**). $S_L$ and $S_V$ are the expansion/ contraction ratio in length and in volume of the hydrogel, respectively, with the reference state at room temperature. Scale bars in **b** and **c** are 1 mm; scale bar in **f** is 5 mm.

that along the ⊥ direction (Supplementary Fig. 9). The specific mechanical properties result from the anisotropic structures composed of cofacially aligned NSs and flexible PNIPAAm network. Rigid NSs with a large aspect ratio act as stiffeners against in-plane deformation, leading to the larger modulus along the // direction[18]. The hydrogel also shows optical anisotropy (Supplementary Fig. 10) and anisotropic response to external stimulus (Fig. 1f). After being transferred into hot water (40 °C), the gel readily becomes opaque and expands to 1.5 times in the ⊥ direction, and contracts to 0.8 of the original length in the // and *n* directions within 15 s (Fig. 1g). During this process, the hydrogel's volume is almost unchanged. This isochoric deformation arises from the rapid dehydration of PNIPAAm above the lower critical solution temperature (LCST), $T_c$, and the increase in

electrostatic permittivity of the media with more free water molecules[21,37]. The sudden increase in permittivity leads to increased electrostatic repulsion between neighboring NSs and thus anisotropic deformation of the gel. As a control, a PNIPAAm gel without NS becomes opaque in hot water, yet exhibits almost no dimension change within 30 s. The unchanged volume indicates no water release from the gel within a short time, and the isochoric deformation is solely driven by temperature-mediated variation of permittivity. Such fast isochoric deformation of the monodomain hydrogel is fully reversible upon cyclic immersion in 25 and 40 °C water baths (Fig. 1h). However, the volume of a PNIPAAm gel ultimately decreases after long-term storage at 40 °C, with or without NS. The linear contraction of monodomain hydrogel is 0.7 and 0.5 of

the original length in the ⊥ and // directions, respectively. The long-term behavior of the PNIPAAm hydrogel is enabled by the long-range migration of water molecules out the gel matrix, and is thus limited by the speed of diffusion and depends on the sample size. For the samples used in the current work, volume contraction of the gel is evident after incubation at 40 °C for ~1 min and fully completes after ~2 h.

**Photo-responsiveness and programmed deformations.** Although the temperature-triggered isochoric deformation of the monodomain hydrogel is fast, there are still two difficulties hindering swift locomotion of the hydrogel: (i) changing the environmental temperature is too slow for effective switching; (ii) selective local actuation is required for programmable locomotion, yet difficult to achieve by direct heating. To address these issues, AuNPs as the photothermal agent are incorporated into the hydrogel to afford photo-responsiveness (Fig. 2a)[38,39].

The presence of AuNPs does not influence the electrical orientation of NSs or the mechanical properties of resultant hydrogel (Supplementary Fig. 11). The composite hydrogel in pink color shows a characteristic absorption peak of AuNPs at 520 nm (Fig. 2b). Owing to the high photothermal efficiency of AuNPs, the local temperature of the hydrogel can be modulated by light irradiation with a high spatiotemporal resolution. The speed and amplitude of temperature rise depend on the power intensity and continuance of irradiation (Fig. 2c). The local temperature of the gel quickly rises from 27 to 55 °C within 3 s under the irradiation of green light at an intensity of 4.5 W cm$^{-2}$. The hydrogel shows reversible isochoric deformation by switching on and off the light irradiation (Fig. 2d and Supplementary Movie 1). The correlation between the changes in the local temperature and the gel dimensions is shown in Fig. 2e. Repeated light irradiation leads to controllable temperature modulation across the $T_c$ of PNIPAAm and thus reversible isochoric deformation of the hydrogel at a fast speed.

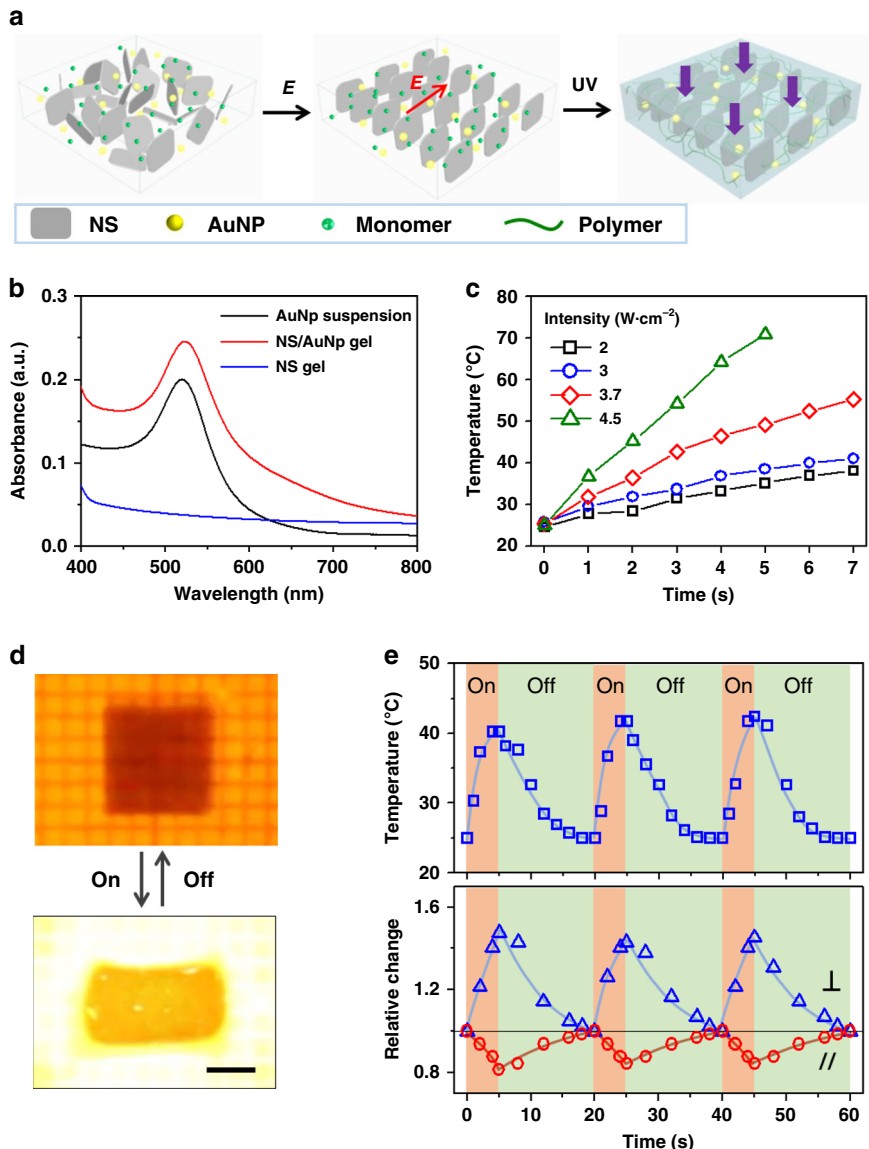

**Fig. 2 Photo-thermo response of the AuNP-containing anisotropic hydrogel. a** Schematic for the fabrication of anisotropic hydrogel after incorporating AuNPs into the precursor solution. **b** Absorption spectra of the aqueous suspension of AuNPs, anisotropic hydrogels with and without AuNPs. **c** Variations of localized temperature of the gel under the irradiation of 520 nm green light with different power intensity. **d**, **e** Photos (**d**) and variations of the localized temperature and gel dimensions (**e**) to show the reversible anisotropic deformation of the AuNP-containing hydrogel under cyclic irradiation of green light with the intensity of 3.7 W cm$^{-2}$. Scale bar in **d** is 1 cm.

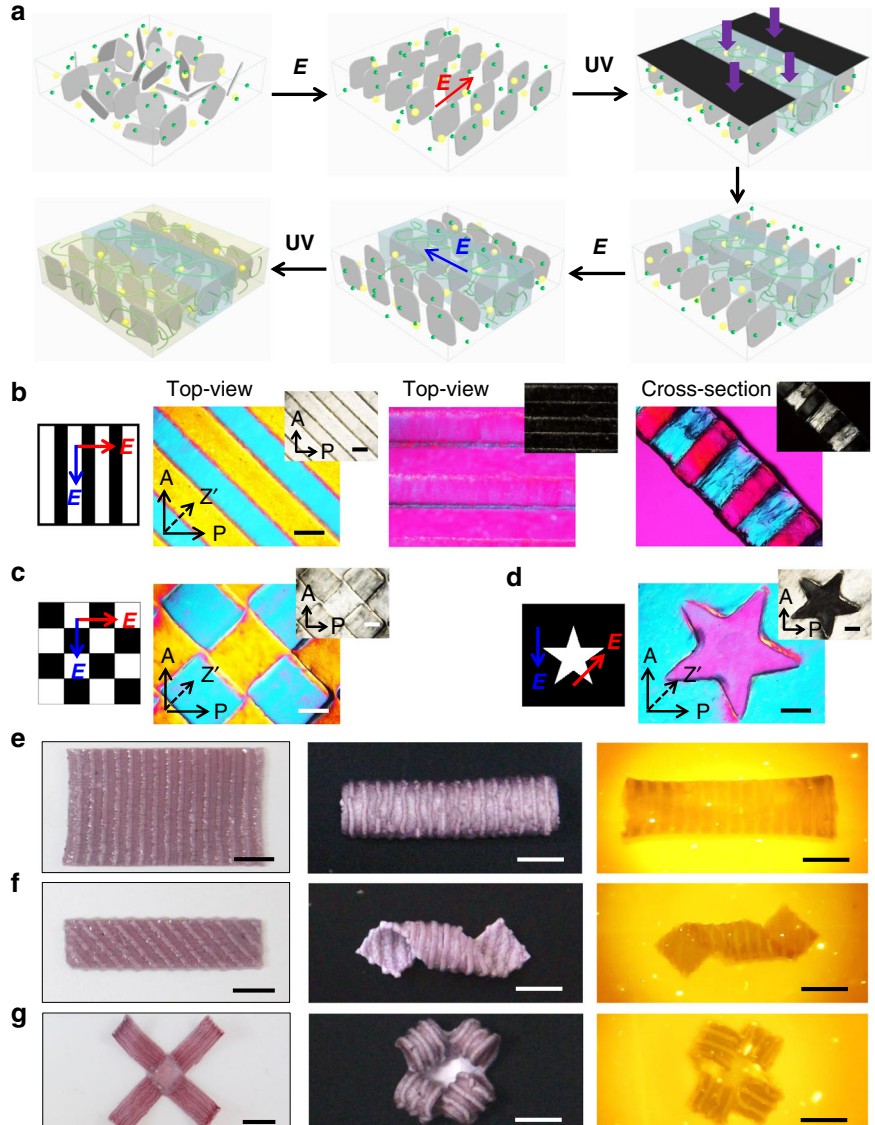

**Fig. 3 Multi-step fabrication and programmed deformation of patterned anisotropic hydrogels. a–d** Schematic for the preparation of patterned anisotropic hydrogel containing AuNPs by multi-step electrical orientation and photolithographic polymerization (**a**) and POM images of representative hydrogels patterned with different masks and alignment of NSs (**b–d**). Scale bar, 1 mm. **e–g** Photos of patterned anisotropic hydrogels before (the left column) and after the actuation of shape changing by incubating in a 40 °C water bath (the middle column) or by remote light irradiation (the right column). A cut-off filter is equipped on the camera to filter out the strong green light for better visualization of the hydrogel under the light irradiation. Scale bar, 5 mm.

Robotic locomotion usually relies on shape changing, which in turn is driven by the build-up of internal stress and is related to the control of structural heterogeneity[40–42]. Hydrogels with patterned orientations of NSs are fabricated by multi-step electrical orientation and photolithographic polymerization, as schemed in Fig. 3a. For example, the stripe-patterned hydrogel with orthogonal orientations of NSs shows birefringence colors on the cross-section with alternating dark and bright regions, indicating the alignments of NSs along the electric field in each step (Fig. 3b). Other patterned hydrogels with different superstructures can also be facilely developed by programming the sequential applications of electric field and photo-polymerization with different masks (Fig. 3c, d). For example, the checkerboard-patterned hydrogel shows birefringence colors, when the orthogonal alignments of NSs are ±45° relative to the polarizer. However, no birefringence is observed in the same hydrogel after a rotation of 45°.

In the patterned hydrogel, discrete regions deform along different directions under external stimulus. The deformation mismatch between neighboring regions leads to the build-up of internal stresses[43], driving the buckling of the hydrogel to form 3D configurations (Fig. 3e–g). Here, the stripe-patterned hydrogel is chosen as a proof of concept. When the stripes are patterned at an angle, θ = 45°, relative to the longitudinal direction of the gel, the internal stresses along the stripes result in buckling upward or downward to form left- or right-handed helix in hot water (Fig. 3f). When θ = 90° or 0°, the patterned hydrogel sheet deforms into a tubular or rolled shape (Fig. 3e and Supplementary Fig. 12). The internal stresses and final 3D configurations of the composite hydrogels can be facilely tailored by the manipulation of electric field and the design of mask pattern (Fig. 3g). The presence of AuNPs in those patterned hydrogels enables the light-triggered reversible deformations in a more flexible manner. Such

programmable deformations lay the foundation for the subsequent design of soft robots.

**Light-steered multi-gait locomotion and its kinematics**. Dynamic control of separate regions in a coordinated fashion is crucial for the programmed locomotion of soft robots[24]. In contrast to flying or swimming that relies on vibration or undulation to exert forces on the environment[24,44], the translational motion of a soft robot on a solid surface requires dynamic manipulation on the local friction underneath to generate propulsion[1,45–47]. Here, this challenging task is neatly completed through spatiotemporally selective actuation of the muscle-like hydrogel by using a moving light beam.

A slender monodomain hydrogel sheet is placed on a polyvinyl chloride (PVC) substrate and actuated by repeated unidirectional scanning of a light beam to mimic the peristaltic crawling of a snail. When the NSs are aligned longitudinally, the rectangular hydrogel crawls rhythmically forward along the scanning direction (Fig. 4a and Supplementary Movie 2). The localized light irradiation leads to longitudinal extension of the gel. Consequently, the moving light results in traveling extension deformation and crawling forward of the gel (Fig. 4c). By contrast, when the NSs are aligned laterally, the gel sheet shows traveling contraction and crawls backward, opposite of the scanning of light (Fig. 4b, c and Supplementary Movie 3). The crawling speed of this gel depends on gel dimensions, power intensity, and scanning speed of the light (Supplementary Fig. 13).

Besides the traveling deformation, another key point for crawling is the modulation of friction force that converts shape changing into directional motion. When we carefully examine the motion of the gel relative to the substrate, the non-slip point is always below the irradiated spot instead of the barycenter. This phenomenon directs our attention to the adhesion and friction between the gel and the substrate after light irradiation. By measuring the friction coefficient, $\mu$, of a PNIPAAm hydrogel against a PVC substrate[48,49], we find that the gel at temperature above $T_c$ has relatively high value of $\mu$ due to the enhanced hydrophobicity (Fig. 4d). For AuNP-containing hydrogel, $\mu$ increases readily and decreases gradually after switching on and off the light irradiation, respectively (Fig. 4e). It is rational that the gel sticks on the hydrophobic PVC substrate upon light irradiation and is set free after removal of the light. The irradiated region has a temperature-induced strain along the axial direction, and the part ahead is heated and deforms before becoming the next anchoring point. The synchronized traveling deformation and anchoring account for the crawling (Supplementary Fig. 14). On the other hand, by introducing a temperature-dependent friction coefficient, our numerical simulations further confirm such a mechanism (Fig. 4f, g and Supplementary Movies 4 and 5). As a reference, when the same experiment is carried out over a hydrophilic hydrogel, a traveling deformation is observed, but the monodomain hydrogel shows no displacement due to negligible difference in friction (Supplementary Movie 6). Even reverse crawling of the gel is observed on a hydrophilic glass substrate, opposite to that on the PVC substrate (Supplementary Movie 7). This is because $\mu$ of the gel against glass substrate is relatively small at high temperature (Supplementary Fig. 15), and the gel moves backward due to the synchronized traveling deformation and detachment (Supplementary Fig. 14).

As a second proof-of-concept example, a stripe-patterned hydrogel is used to mimic the walking gait of inchworm. Owing to the heterogeneous structure and swelling mismatch, the patterned hydrogel is slightly curved when equilibrated in water at room temperature[41]. Localized irradiation dramatically increases the internal stress and thus the bending curvature. The spatiotemporal light directs the traveling bending deformation, as well as the dynamic friction on the two "feet", resulting in walking of the hydrogel opposite of the scanning direction. The structural symmetry allows the walking direction to be reversed simply by switching the scanning direction (Fig. 5a, b and Supplementary Movie 8). The coordinated sequence of deformation and friction force is manifested by the discrete displacements of the two "feet" at distinct period of time (inset of Fig. 5c), favoring highly efficient walking. The stride is ~6 mm per cycle of light scanning. The detailed walking kinematics is examined through numerical simulation (Fig. 5d and Supplementary Movie 9). When the light scans from left to right, the left end of hydrogel is heated first and used as the standing foot while the gel bends further with the motion of light beam, dragging the other foot closer. The right end is heated with the approaching of the light beam and becomes the standing foot, after which the relaxation of the hydrogel moves the left foot further left. As expected, the walking stride depends on the gel dimensions and scanning conditions (Supplementary Fig. 16).

When the scanning path of the laser beam is at an angle to the stripes, the patterned hydrogel changes its orientation after each scanning (~11° per stroke when scanned along the diagonal) (Fig. 5e–g and Supplementary Movie 10). Sequential scanning can accumulate the rotation (Supplementary Movie 11). Gel dimensions and scanning conditions also influence the turning stride (Supplementary Fig. 16). Simulations further reveal the turning kinematics by considering the coordinated sequence of localized deformation and dynamic friction of the gel (Fig. 5h and Supplementary Movie 12). When the light beam enters from the top-left corner, the gel bends more on the upper part while the left edge sticks to the substrate. The nonuniform bending twists the gel, turns the right edge counterclockwise, and brings the top-right corner closer to the left. Later when the light spot moves along the diagonal to the bottom-right, the local heating anchors the already tilted right edge and releases the left edge, along with the recovery of the hydrogel to a relaxed state in the tilted orientation. Such synergistic shape-changing and stick-slip transition realize the turning motion of the gel.

These examples clearly demonstrate the biomimetic locomotion realized by harnessing the rapid isochoric deformation of the muscle-like hydrogel and the mutual coordination of shape-morphing and dynamic friction. The locomotion speed is comparable to or higher than that of other hydrogel-based soft robots which rely on ratchet substrates, asymmetrical shapes, or geometric confinements (Supplementary Table 1); the locomotion of our continuum soft robots can be speeded up further by increasing the loading content of AuNPs, power intensity, and scanning speed of the light. More sophisticated motions can be accomplished by tailoring the design of hydrogel-based soft robots (e.g., the patterns and local orientations) and the control of light (e.g., the structure and spatiotemporal manipulation).

## Discussion

Based on the spatiotemporal actuation of muscle-like hydrogels that response anisotropically to light at a high speed and a large amplitude, we have developed a design of soft robots with the capability of versatile locomotion. By multi-step electrical orientation and photolithographic polymerization, patterned hydrogels showing programmed deformations into 3D configurations are fabricated. Sophisticated biomimetic motions, including crawling, walking, and turning, are realized in the hydrogel-based robots by using a light beam to spatiotemporally actuate the shape morphing and dynamically control the friction against the substrate. Other versatile motion gaits can be realized by designing

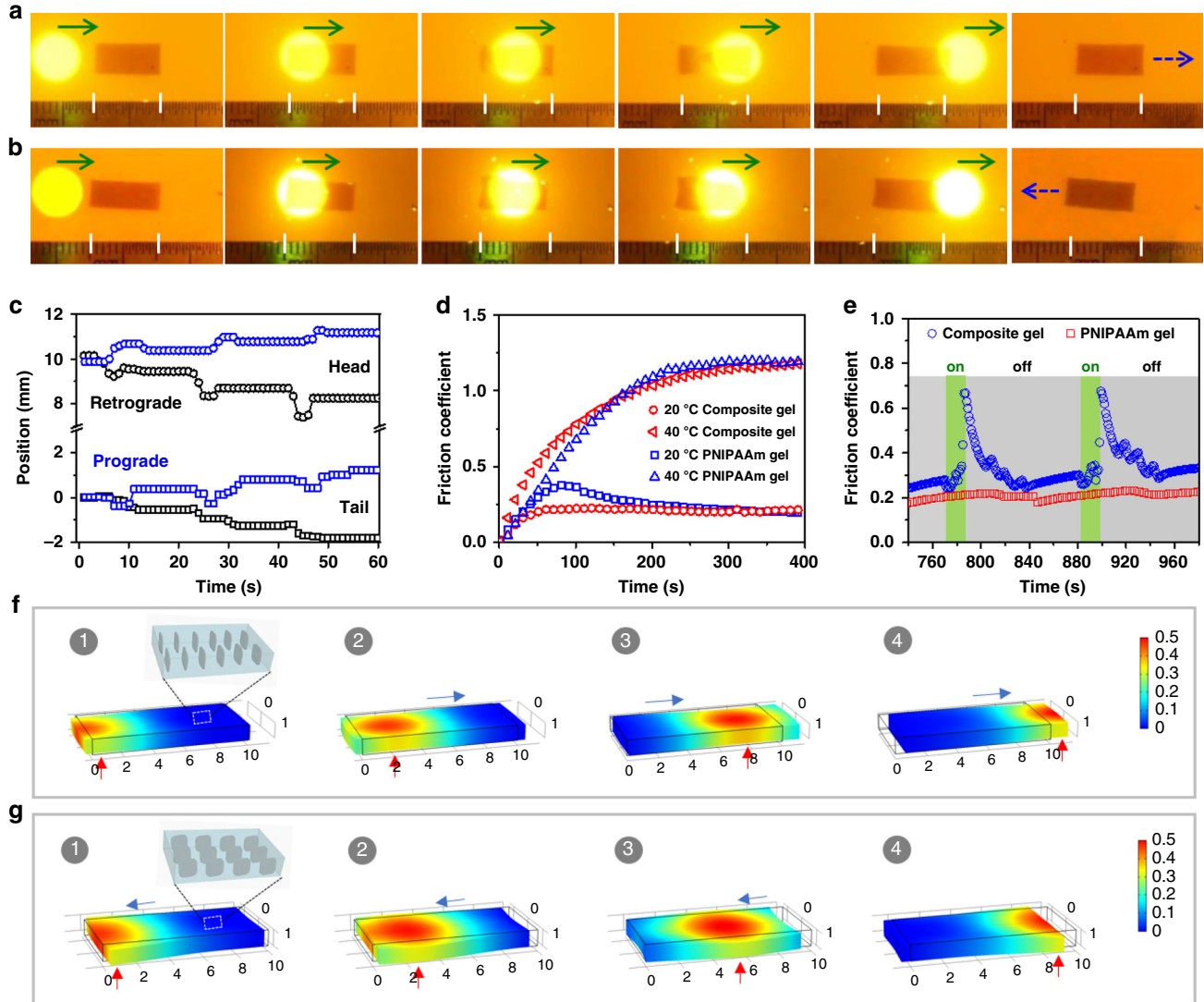

**Fig. 4 Photo-steered crawling of the monodomain hydrogels. a, b** Snapshots to show the crawling motions of the slender hydrogel sheets with alignment of NSs along (**a**) or perpendicular (**b**) to the long axis under the scanning of a light beam. A cut-off filter is applied to filter out the strong green light for better visualization of the hydrogel under light irradiation. Green and blue arrows indicate the light scanning direction and motion direction, respectively. Gel dimensions, 10 mm × 5 mm × 1 mm; power intensity, 3.5 W cm$^{-2}$; scanning speed, 1.7 mm s$^{-1}$. **c** Time-variant displacements of the head and the tail of the slender gel sheets under cyclic scanning of a light beam. **d, e** Variations of the friction coefficient of the PNIPAAm hydrogel and the NS/AuNP-containing PNIPAAm hydrogel against a PVC substrate as a function of test time at 20 and 40 °C (**d**) or under cyclic irradiation of a green-light beam at 20 °C (**e**). **f, g** Simulation results to show the kinematics of the forward (**f**) and retrograde (**g**) crawling motions of the monodomain hydrogel on the basis of photo-directed localized isochoric deformation and dynamic friction modulation. The red and blue arrows indicate the anchoring spot and motion direction of the gel. The color scale indicates the normalized temperature.

the hydrogel and manipulating the light to achieve dynamic synergy at specific regions. The working mechanism allows the robots to be scaled down to micrometer size for biomedical and engineering applications[50,51]. The structural design of muscle-like hydrogels and the light-enabled spatiotemporal manipulation demonstrated here should also be informative for developing other soft robots with advanced technologies and versatile applications.

## Methods

**Materials**. *N*-isopropylacrylamide (NIPAAm) was used as received from Aladdin Chemistry Co., Ltd; *N,N′*-methylenebis(acrylamide) (MBAA, used as the chemical crosslinker) was purchased from Sigma Aldrich. Lithium phenyl-2,4,6-tri-methylbenzoylphosphinate (LAP, used as the photoinitiator) was synthesized according to the reported protocol[52]. Fluorohectorite $Na_{0.5}[Li_{0.5}Mg_{2.5}][Si_4]O_{10}F_2$ NSs were prepared by melt synthesis at high temperature well above 1000 K

followed by a long-term annealing process[53]. AuNPs were synthesized according to the reported method[54]. Millipore deionized water was used in all the experiments.

**Electric-field-induced orientation of NSs**. Aqueous suspensions of the NSs were prepared by addition of a prescribed amount of water to the fluorohectorite $[Na_{0.5}]$ $[Li_{0.5}Mg_{2.5}][Si_4]O_{10}F_2$ powder. The mixture was oscillated with a speed of 60 r.p.m. for 3 months at 25 °C, which led to repulsive osmotic delamination of the fluor-ohectorites into single-layered NSs in the suspensions. The aqueous suspension was injected into a cell consisting of a pair of parallel glasses with 2 mm silicon spacer, and an AC electric field with high frequency was applied to the cell by using a pair of Ag electrodes with a certain distance placed in the suspension. After action of the electric field for a certain period of time, the NSs were aligned along the direction of the electric field.

**Synthesis of monodomain nanocomposite hydrogels**. The precursor solution was prepared by dissolving a prescribed amount of NIPAAm, MBAA, and LAP in the homogeneous aqueous suspension (0.27 wt%) of NS (Supplementary Table 2). After injecting the precursor solution into the reaction cell consisting of a pair of

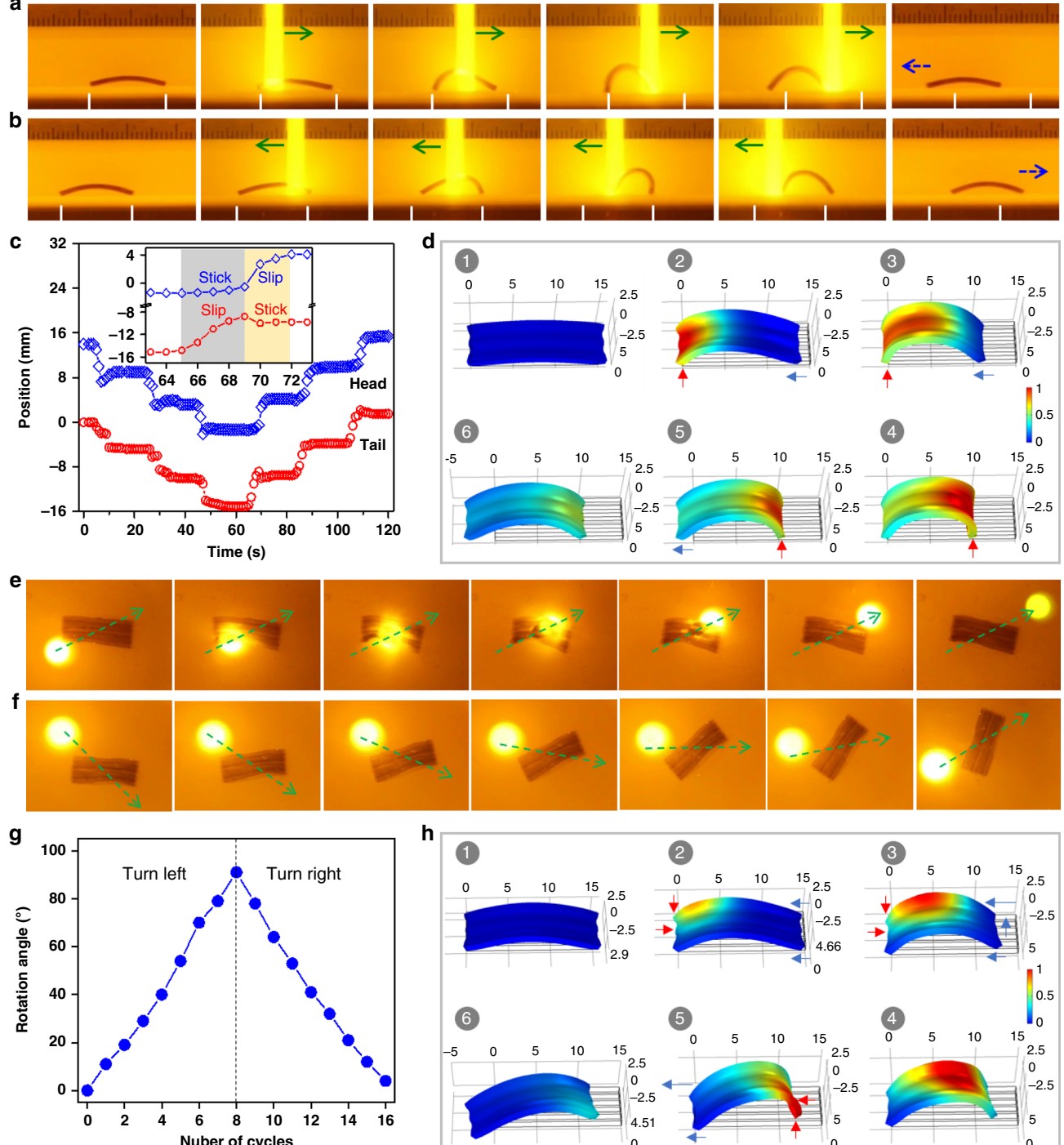

**Fig. 5 Photo-steered walking and turning of the patterned anisotropic hydrogel. a**, **b** Snapshots to show the walking gait of the stripe-patterned anisotropic hydrogel under the scanning of a light beam from left to right (**a**) and from right to left (**b**). Green and blue arrows indicate the light scanning direction and motion direction, respectively. **c** Displacements of the two ends (heat and tail) of the gel under cyclic scanning of light. **d** Simulation results showing the kinematics of the retrograde walking of the hydrogel based on photo-directed buckling and dynamic adhesion. **e**–**g** Snapshots to show the turning motion in one cycle of light scanning (**e**), and the sequential turning motions (**f**) and accumulated rotated angle (**g**) of the pattern hydrogel under cyclic scanning of a light beam along the diagonal direction of the gel. Green arrows indicate the light scanning directions. **h** Simulation results showing the kinematics of the turning motion of the hydrogel under the scanning of light. The red and blue arrows indicate the anchoring spot and motion direction of the gel. Color scale in the simulated results indicates the normalized temperature. Gel dimensions, 15 mm × 5 mm × 1 mm; power intensity, 3.5 W cm$^{-2}$; scanning speed, 1.7 mm s$^{-1}$.

parallel glasses with 1 mm silicon spacer, an AC electric field with high frequency was applied to the cell by using a pair of Ag electrodes with 20 mm distance placed in the solution. After a period of time, the reaction cell containing precursor solution with electrically oriented NSs was immediately exposed to UV light irradiation for 30 s to initiate the polymerization and crosslinking to obtain monodomain nanocomposite hydrogel. To prepare photo-responsive hydrogel, a prescribed amount of AuNPs (0.013 wt%) was incorporated into the precursor solution described above, followed by electric-field-induced orientation and photo-polymerization. The resultant hydrogels were swelled in a large amount of water for several days to remove the residuals and achieve the equilibrium state.

**Synthesis of patterned anisotropic hydrogels**. Patterned anisotropic hydrogels with complex ordered structures were fabricated by multi-step electric-field-induced orientation of NSs and photolithographic polymerization. After the electric-field-induced orientation of NSs in the precursor solution, the reaction cell was exposed to UV light irradiation for 30 s through a photomask prepared by ink-jet printing. Reaction took place at the UV-exposed regions to form anisotropic gels, whereas there was no reaction at the protected regions where the NSs can be reoriented under another electric field. After resetting the position of Ag electrodes, a distinct electric field with a specific direction was applied to the sample for 1 h. Then, the sample was exposed to UV light for 30 s without a mask, resulting in polymerization of the precursor solution in the previously protected regions to freeze the reoriented structure of NSs. The obtained as-prepared patterned hydrogel with complex ordered structures of NSs was swelled in a large amount of water for several days to remove the residuals and achieve the equilibrium state.

**Deformations and locomotion of the hydrogels**. The thermo- and photo-responsive processes of the monodomain composite hydrogel were recorded by a digital camera and then analyzed by measuring the variations of gel dimensions. The variations in length ($S_L$) (i.e. along with the alignment of NS), width (i.e. perpendicular to the alignment of NS), and thickness of the monodomain hydrogel are calculated as $S_L(//) = L_1/L_0$, $S_L(\perp) = W_1/W_0$, and $S_L(n) = T_1/T_0$, respectively, in which, $L$, $W$, and $T$ are the length, width, and thickness of the sample and the subscript numbers 1 and 0 correspond to the deformed state and the original equilibrated state, respectively. The dimensions of hydrogels were analyzed from the snapshots of a movie that recorded the fast shape deformation of the gel after being transferred from 25 °C water bath into 40 °C water bath or being directly irradiated with 520 nm green light (intensity, 5.4 W cm$^{-2}$) at room temperature. The swelling ratio in volume ($S_V$) of the hydrogel was calculated by $S_V = S_L(//) \cdot S_L(\perp) \cdot S_L(n)$. The variations of swelling ratio in length ($S_L$) and in volume ($S_V$) of pure PNIPAAm gel were also investigated for comparison. Deformations of the patterned composite hydrogels were triggered by incubating the gels in hot water (40 °C) or by photoirradiation (520 nm green light). The configurations were recorded by using a digital camera with a cut-off filter (550–1100 nm) to filter out the strong green light.

Scanning of a green light beam (spot diameter: 7 mm) along a specific direction was applied to power and steer the robotic locomotion of the anisotropic composite hydrogel on a PVC surface. Crawling motions of the slender hydrogel sheets (dimensions: 10 mm × 5 mm × 1 mm) with alignment of NSs along or perpendicular to the long axis were realized under the cyclic scanning of a light beam with a certain intensity through the hydrogel at proper direction and rate. Walking and turning motions of the stripe-patterned hydrogel were realized under the cyclic scanning of a light spot with different intensities and scanning rates along the long axis and the diagonal of the rectangular hydrogel sheet (dimensions: 15 mm × 5 mm × 1 mm). The locomotion was recorded as the movie by using a digital camera with a cut-off filter (550–1100 nm) to filter out the strong green light.

**Characterizations**. The birefringence of NS suspensions and resultant anisotropic hydrogels was observed under a POM (LV100N POL, Nikon) with and without a 530 nm tint plate. The as-prepared and equilibrated gels were sliced into narrow strips for the observation from different directions. SAXS measurements were performed to the monodomain hydrogel on the BL16B beamline with an X-ray wavelength of 0.124 nm at Shanghai Synchrotron Radiation Facility. The beam spot was 172 × 172 μm$^2$, and the sample-to-detector distance was 1880 mm. Intensity distribution profile in the azimuthal angle ($\phi$) was used to calculate the orientation index ($\pi$) according to the equation of $\pi = (180 - H)/180$, where $H$ is the half-width of the peak of the azimuthal profile from the selected equatorial reflection.

The mechanical properties of monodomain hydrogels in the equilibrated state were measured at room temperature by using a tensile tester (Instron 3343). The hydrogel sheet was cut into dumbbell-shaped samples, with a gauge length of 12 mm and a width of 2 mm, in the direction parallel and perpendicular to the alignment of NS. Tensile tests were performed with a stretch rate of 100 mm min$^{-1}$. Young's modulus was calculated based on the initial slope of the nominal stress−strain curve with a strain below 10%. Young's modulus ($E$), breaking stress ($\sigma_b$), and breaking strain ($\varepsilon_b$) were obtained from three parallel measurements.

Dynamic friction coefficient of the hydrogel against a substrate was measured by using a DHR-2 rheometer (TA Instruments, USA). The isotropic PNIPAAm hydrogel with and without NSs and AuNPs was equilibrated in water at 20 or 40 °C

before being cut into disc-shaped specimens with a diameter of 20 mm. The specimen was glued to the top plate; the PVC substrate, tough double-network hydrogel sheet (prepared according to the procedure reported in ref. [55]), or glass substrate was glued to the Peltier plate of the rheometer. After setting the temperature of Peltier plate to 20 or 40 °C, a few drops of water were dripped onto the substrate. Then the top plate was lowered to contact with the Peltier plate; the preliminary normal force ($F_{n,0}$) was set as 0.3 N. Shear sweep was performed to the specimen with a constant shear rate of 0.001 s$^{-1}$, the torque ($T$) and normal force ($F_n$) were recorded as a function of time. To examine the friction coefficient of the hydrogel under light irradiation, the PVC sheet was glued to the top plate of the rheometer, and a disc-shaped specimen with a diameter of 8 mm was immobilized on a polymethyl methacrylate rack under which the laser pointer was aligned to the specimen. The 520 nm green light was turned on and off cyclically during the shear sweep. The friction coefficient ($\mu$) was determined by $\mu = F/F_n = 4 T/(3RF_n)$, where $F$ and $R$ are the friction force and specimen's radius, respectively[56].

**Theoretical model**. The dynamic deformation and locomotion of the hydrogels are modeled by solving coupled equations of hyperelasticity and nonlinear friction. Similar as the usual treatment on anisotropic thermal strains[57], the phase-transition-induced isochoric deformation is modeled through an inelastic deformation gradient tensor

$$\mathbf{F}^{(i)} = \lambda_\perp \mathbf{e}_\perp \otimes \mathbf{e}_\perp + \frac{1}{\sqrt{\lambda_\perp}} \mathbf{e}_\parallel \otimes \mathbf{e}_\parallel + \frac{1}{\sqrt{\lambda_\perp}} \mathbf{e}_n \otimes \mathbf{e}_n, \tag{1}$$

where $\lambda_\perp$ is the stretch in the $\perp$ direction, $\mathbf{e}_\perp$, $\mathbf{e}_\parallel$, and $\mathbf{e}_n$ are the unit vectors along the corresponding directions, and $\otimes$ represents a tensor product. Knowing that the deformation is relatively fast compared to the heating rate, it is further assumed that $\lambda_\perp$ ramps linearly from the normalized temperature $T = 0$ (room temperature) to $T = 1$ at which the isochoric deformation saturates.

By using a multiplicative decomposition, we write the deformation gradient tensor into an inner product between the inelastic and elastic deformation gradients[58]

$$F = \mathbf{F}^{(e)} \cdot \mathbf{F}^{(i)}, \tag{2}$$

and express the Helmholtz free energy density $W$ in terms of $\mathbf{F}^{(e)}$. It is believed that the observed phenomena are dominated by the isochoric deformation of the gel, and is less dependent on its anisotropic elastic properties. For simplicity, we adopt a neo-Hookean model with isotropic initial shear modulus $G$

$$W(\mathbf{F}^{(e)}) = \frac{G}{2}(\mathbf{F}^{(e)} : \mathbf{F}^{(e)} - 3), \tag{3}$$

and assume volume incompressibility. The equation of state can then be given in terms of the nominal stress

$$s_{iK} = \frac{\partial W}{\partial F_{iK}}, \tag{4}$$

which satisfies the equilibrium equation

$$\nabla \cdot s = 0, \tag{5}$$

in the bulk, and boundary condition $\mathbf{s} \cdot \mathbf{N} = \mathbf{t}$ on a surface of prescribed nominal traction $\mathbf{t}$, where $\mathbf{N}$ is the unit normal vector of the surface. As the motion of the gels is relatively slow, all inertial effects are neglected.

With the aid of a fast heating laser beam, the region of elevated temperature is usually localized with a minor effect of heat conduction. We thus simplify the temperature evolution by assuming a modified Stefan–Boltzmann equation

$$\frac{\partial T(\mathbf{X},t)}{\partial t} = r(\mathbf{X}, t) - \beta T(\mathbf{X}, t)^4. \tag{6}$$

Here $r(\mathbf{X}, t)$ characterizes the power of the laser beam, which is localized at the irradiated region, and $\beta$ is related to the emissivity of the gel to the environment. Both parameters are obtained by fitting the dynamic response of a monodomain gel (Fig. 1h).

The effect of friction is modeled as a rate-dependent tangential force over the surface of contact

$$f = -\eta(T)v, \tag{7}$$

where $\mathbf{v}$ is the relative velocity, and the viscous coefficient $\eta(T)$ is taken to be a smoothed step function of temperature that increases by a factor of 10 near the LCST of PNIPAAm. As the frictional force over the entire contact surface is self-balanced, neither the actual value of the viscous coefficient nor the weight of the gel samples affects the kinematics.

Upon substitution of Eqs. (1)–(4) and (7), Eqs. (5) and (6) can be solved numerically by using a finite element method through the commercial package COMSOL Multiphysics 5.4. Three-dimensional models representing the actual geometries of the samples are developed. Linear tetrahedral elements are used to interpolate the field of displacement, and the normalized temperature is evolved over the Gauss points. To capture the inhomogeneous deformation of the stripes in different orientations, a mesh size of ~0.1 mm is taken in the transition zones.

## Data availability
The data that support the findings of this study are available from the corresponding authors upon reasonable request.

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

## Acknowledgements

This research is financially supported by National Natural Science Foundation of China (No. 51773179 and 51973189) and Natural Science Foundation of Zhejiang Province of China (No. LR19E030002).

## Author contributions

Q.L.Z. and Z.L.W. convinced this concept and designed the experiments. Q.L.Z., C.D., and Y.D. performed the experiments with the assistance of M.D. M.M., and J.B. for the synthesis and characterization of nanosheets. W.H. carried out theoretical and numerical analyses. Q.L.Z., C.D., J.B., W.H., Z.Q., and Z.L.W. contributed to the discussion and interpretation of the results. Q.L.Z., W.H., and Z.L.W. wrote the paper. Z.L.W. supervised the research.

## Competing interests

The authors declare no competing interests.
