## [Peer Review File · Nature Communications]

Reviewers' Comments:

Reviewer #1:

Remarks to the Author:

The authors have addressed all my raised points from the previous review very well, and I therefore recommend the article for publication in *Nature Communications*. The article reads well with the added corrections, and they have been implemented in a careful and consistent way.

Reviewer #2:

Remarks to the Author:

The submission contributed by Wu et al reports the preparation of hydrogel photoactuators that show anisotropic motion under the light irradiation. Electrical field induced alignment of nanosheets in hydrogel matrix enable the anisotropic contraction of the hydrogels, and photothermally driven actuation is achieved by incorporation of gold nanoparticles. The authors clearly demonstrate the programmable locomotion of the hydrogels in response to the scanning of the green light. Even though some concepts of the report (alignment of nanosheet in the gel, photothermal volume control, and patterned crosslinking) is already extensively reported, the authors smartly combined these concepts with their own ideas. However, the following points should be revised or discussed to recommend the acceptance of the publication.

1. Compared with the previous reports regarding the locomotion of the hydrogels, the authors should discuss the actuation speed.
2. page 4, line 6: following sentence is hard to understand. "After the external field is switched off, the alignment is well maintained within 8 min, but deteriorates gradually after long-term storage due to structural relaxation"
Does this sentence mean the aligned nanosheet in the hydrogel are relaxed during long-term storage of the hydrogel? If so, how to guarantee the durability of the actuators?
3. Alignments of the nanosheet in the gel yields the anisotropic mechanical properties, but a detail discussion of the reason is not given in the manuscript.

Response to Reviewers' Comments

Reviewer #1 (Remarks to the Author):

The authors have addressed all my raised points from the previous review very well, and I therefore recommend the article for publication in Nature Communications. The article reads well with the added corrections, and they have been implemented in a careful and consistent way.

Response: We sincerely appreciate for the recommendation for publication of our manuscript.

Reviewer #2 (Remarks to the Author):

The submission contributed by Wu et al reports the preparation of hydrogel photoactuators that show anisotropic motion under the light irradiation. Electrical field induced alignment of nanosheets in hydrogel matrix enable the anisotropic contraction of the hydrogels, and photothermally driven actuation is achieved by incorporation of gold nanoparticles. The authors clearly demonstrate the programmable locomotion of the hydrogels in response to the scanning of the green light. Even though some concepts of the report (alignment of nanosheet in the gel, photothermal volume control, and patterned crosslinking) is already extensively reported, the authors smartly combined these concepts with their own ideas. However, the following points should be revised or discussed to recommend the acceptance of the publication.

Response: Thank you very much for the positive evaluation and recommendation for publication of our work.

1. Compared with the previous reports regarding the locomotion of the hydrogels, the authors should discuss the actuation speed.

Response: According to the constructive suggestion, we have added some discussions about the locomotion speed and compared it with the values of gel-based soft actuators/robots reported in the literatures (page xx and Supplementary Table Sx).

2. page 4, line 6: following sentence is hard to understand. "After the external field is switched off, the alignment is well maintained within 8 min, but deteriorates gradually after long-term storage due to structural relaxation"

Does this sentence mean the aligned nanosheet in the hydrogel are relaxed during long-term storage of the hydrogel? If so, how to guarantee the durability of the actuators?

Response: After polymerization, the structure of nanosheets is fixed in the gel matrix and does not change over time. The sentence is used to state the variations of aligned nanosheets in the precursor solution after the electric field is switched off, prior to polymerization. As shown in Supplementary Figure S6, the anisotropic structure of the nanosheets in the precursor solution is well maintained in the first 8 min, yet

gradually deteriorates after a long period of storage due to the thermal fluctuation-induced structural relaxation of the nanosheets. To make this point clear, we revised the sentence to “After the external field is switched off, the alignment of NSs in the precursor is well maintained within 8 min, but deteriorates gradually after long-term storage due to structural relaxation (Supplementary Fig. 6).” The precursor solution state of the sample is also described in the caption of Supplementary Fig. 6.

3. Alignments of the nanosheet in the gel yields the anisotropic mechanical properties, but a detail discussion of the reason is not given in the manuscript.

Response: The specific mechanical properties of the monodomain hydrogel result from the anisotropic structures composed of cofacially aligned NSs and flexible PNIPAAm network. Due to the large aspect ratio and rigidity of NSs, the local deformation in the gel parallel to each NS is constrained, while that normal to each NS is unconstrained. Such a difference induces the larger stiffness in-plane with the aligned NSs, and low stiffness in the perpendicular direction. This discussion has been added to revised manuscript (pages 4 and 5).